# Quantitative Parameters Relevant for Diabetic Macular Edema Evaluation by Optical Coherence Tomography Angiography

**DOI:** 10.3390/medicina59061120

**Published:** 2023-06-10

**Authors:** Alina-Simona Lazăr, Horia T. Stanca, Bogdana Tăbăcaru, Ciprian Danielescu, Mihnea Munteanu, Simona Stanca

**Affiliations:** 1Doctoral School, “Carol Davila” University of Medicine and Pharmacy, Strada Dionisie Lupu No. 37, 020021 Bucharest, Romania; 2Clinical Department of Ophthalmology, “Prof. Dr. Agrippa Ionescu” Emergency Hospital, Strada Ion Mincu No. 7, 011356 Bucharest, Romania; 3Department of Ophthalmology, Faculty of Medicine, University of Medicine and Pharmacy “Grigore T. Popa”, Strada Universitatii No. 16, 700115 Iasi, Romania; 4Department of Ophthalmology, “Victor Babes” University of Medicine and Pharmacy, 300041 Timisoara, Romania; 5Clinical Department of Pediatrics, University of Medicine and Pharmacy “Carol Davila”, Strada Dionisie Lupu No. 37, 020021 Bucharest, Romania

**Keywords:** diabetic macular edema, optical coherence tomography angiography, vessel density, perfusion density, FAZ, fractal dimension

## Abstract

Diabetic macular edema (DME) is one of the main ocular complications of diabetes mellitus (DM) that can lead to important vision loss in diabetic patients. In clinical practice, there are cases of DME with unsatisfying treatment responses, despite adequate therapeutic management. Diabetic macular ischemia (DMI) is one of the causes suggested to be associated with the persistence of fluid accumulation. Optical coherence tomography angiography (OCTA) is a non-invasive imaging modality, able to give in-depth information about retinal vascularization in a 3-dimensional manner. The OCTA devices currently available can provide various OCTA metrics that quantitatively assess the retinal microvasculature. In this paper, we reviewed the results of multiple studies that investigated the changes in OCTA metrics in the setting of DME and their possible contribution to the diagnosis, therapeutic management, follow-up and prognosis of patients with DME. We analyzed and compared relevant studies that investigated OCTA parameters related to changes in macular perfusion in the setting of DME and we evaluated the correlations between DME and several quantitative parameters, such as vessel density (VD), perfusion density (PD), foveal avascular zone (FAZ)-related parameters, as well as complexity indices of retinal vasculature. The results of our research showed that OCTA metrics, evaluated especially at the level of the deep vascular plexus (DVP), are useful instruments that can contribute to the assessment of patients with DME.

## 1. Introduction

Diabetic macular edema (DME) is known to be the main cause of vision loss in the working-age category of diabetic patients [1]. Current data estimate that the worldwide prevalence of this condition will rise by 51.9% until the year 2045 [2]. Macular edema is an ocular complication of diabetes mellitus (DM), which results from an imbalance between the accumulation and elimination of fluid in and out of the retina, being the consequence of a multifactorial process that implies oxidative stress, inflammation and blood-retinal barrier dysfunction [3]. Macular edema can develop at any stage of diabetic retinopathy (DR), independently of the DR severity [4]. One of the triggers suggested to be responsible for fluid accumulation at the macular level is ischemia, another frequent and debilitating complication of DR [5]. Fluid accumulation in the setting of ischemia is thought to result from endothelial cell injury, the disruption of tight junctions that form the blood-retinal barrier, and consequent vascular hyper-permeability [5], which is also intensified by the concurrent inflammatory processes associated [5].

In clinical practice, an unsatisfying treatment response was noticed in some cases of patients with DME, suggesting that the association between DME and diabetic macular ischemia (DMI) may decrease the response to the treatment modalities for macular edema, with the persistence of long-standing fluid accumulation [6].

Moreover, one of the most important consequences of DMI, in the presence or absence of other ocular complications of diabetes mellitus, is the decrease in visual acuity which may be progressive and irreversible [7]. When associated with other ocular complications of diabetes, such as macular edema, the vision loss can still persist, even if the macular fluid was successfully resolved [8].

Even though fluorescein angiography (FA) has been the mainstay in imaging the retinal vasculature, its usage is limited to a two-dimensional visualization of the superficial vascular plexus (SVP) in the retina [9].

Optical coherence tomography angiography (OCTA) is a noninvasive modality of investigating the retinal and choroidal circulation, without the necessity of intravenous dye injection [10]. It can provide qualitative and quantitative information, increasingly becoming a very useful tool in clinical practice in terms of diagnosis, follow-up and therapeutic decisions in patients with vascular ocular conditions [10]. Compared to FA, it can provide a three-dimensional image of the fundus, allowing us to individually evaluate each of the retinal plexuses, as well as the choriocapillaris and the choroid [10].

In light of the aspects mentioned above, the assessment of the quantitative microvascular parameters can objectively evaluate the degree of macular ischemia in diabetic patients with DME and may represent a relevant topic regarding the therapeutic decision and the prognostic issues.

The aim of this review was to compile the results of various relevant studies that investigated the changes in OCTA metrics in the setting of DME.

## 2. OCTA Microvascular Evaluation of the Retina

It is well known that the retinal vascularization is dependent on two vascular sources. The inner two-thirds of the retina is supplied by the central retinal artery, whereas the outer third of the retina is supplied by blood coming from the choroid [11]. There are four different capillary plexuses that can be distinguished on OCTA, located at different levels in the retina [12]. Close to the optic nerve and minimally represented in the macula, there is the radial peripapillary capillary plexus (RPCP), located within the retinal nerve fiber layer (RNFL) [12]. The vascularization of the macula is principally composed of three vascular plexuses: superficial, intermediate and deep [12]. Both the SVP and the intermediate capillary plexus (ICP) are disposed as a three-dimensional network [12], whereas the deep capillary plexus (DCP) is disposed in a uniplanar fashion [12].

The foveal avascular zone (FAZ) is a relatively circular area around the fovea which lacks vascularization, delimited by an annulus of vascular elements coming from the three retinal plexuses [11]. FAZ is the main OCTA element indicative of ischemia at the macular level in diabetic patients [7,8], but recent technological advancements enhanced the possibility to calculate many other metrics that could be potential indicators of capillary closure or capillary loss in the macula. Their investigation in the setting of diabetic macular edema could bring important information, especially in the subset of patients that lacks anatomic or functional response at the treatment modalities given.

## 3. The Process of OCTA Metrics Calculation

Many OCTA devices have specific software incorporated, designed to make quantitative evaluations of the macular perfusion. Generally, after the OCTA image acquisition is performed, the grayscale en face OCTA image undergoes two main processes for the quantitative assessment of vascularization: binarization and skeletonization.

Binarization is a process that transforms the original grayscale image into a black-and-white one, based on a threshold value [13]. Algorithms of binarization can be classified into two types: global and local [13]. Global binarization algorithms imply the calculation of a single grayscale threshold for the entire image [13], whereas local binarization consists of an algorithm that calculates different thresholds for different zones of the image [13]. Local binarization algorithms threshold the signal without taking into account the differences in brightness or intensity, but, on the other hand, they may also take into account signal correspondence to “noise” [13]. It was reported that in the case of global binarization algorithms, the quantitative OCTA parameters results were lower compared to local binarization algorithms [14]. The difference between global and local binarization thresholds was found to be significantly influenced by the presence of DME [14].

Skeletonization is another process used for OCTA metrics calculation. Skeletonization is based on reducing the caliber of each vessel to a single pixel, thus not taking into account the capillary diameter when doing the analysis [15]. Skeletonization also allows for a uniform contribution of both large and small vessels in density measurements [15]. Skeletonization is the method of choice to calculate the length of the blood vessel as the total length of perfused vasculature per unit area in a region of measurement [15].

Various quantitative parameters can be measured by means of OCTA. However, at the moment, there is heterogeneity regarding certain aspects such as the nomenclature of the parameters, the methods and algorithms used for their calculation, the units of measurement and the segmentation modalities, being difficult to reach a consensus between the results of different studies [16].

One of the parameters that can objectively be measured by means of OCTA is vessel density (VD). According to some sources, VD is defined as a ratio between the total length of perfused vasculature per unit area in a certain region [16]. However, in some cases, due to the lack of terminology standardization, “VD” is used to describe perfusion density (PD), which is in fact, according to other sources, a different parameter, calculated as a ratio of the total area of perfused vasculature to the unit area in a certain region [16] or pixel of vessels per total pixel of the analyzed surface [14]. A clarification regarding the definition of these parameters is necessary to be made before drawing any conclusions regarding the results of the studies at the moment.

Among other parameters of interest, in patients with DME, there is also the vessel diameter index (VDI), which is calculated as a ratio of total vessel area on a binarized image to total vessel length on a skeletonized image [16].

Furthermore, FAZ-related parameters, which are most investigated related to DR and DME, encompass FAZ area (mm^2^), FAZ perimeter (mm), and FAZ circularity index (CI). The CI reflects how close a certain shape resembles a circle [17] and is calculated after the formula CI = (4π x area)/(perimeter^2^) [17]. A ratio closer to 1.0, which is the circularity index of a perfect circle, means that the shape is circular. If the ratio is closer to 0, then the shape is irregular [17]. Among the indices that measure retinal vascular complexity, there is fractal dimension (FD), which shows the branching distribution of the vessels, vascular tortuosity index (VTI) which expresses the curvature of the retinal vessels, and vascular complexity index (VCI) which measures the complex morphology of the retinal vasculature [18].

## 4. Quantitative OCTA Assessment in Healthy Individuals

Regarding the OCTA variability of the microvascular metrics between different retinal capillary plexuses in healthy adults, it was reported that the mean VD values were significantly higher in DCP compared to SCP in all age groups [19]. Moreover, there was a correlation between the age range and the sex-related to VD. It resulted that the variability of VD increased with age, and VD was higher in women than in men after 60 years old [19]. FAZ area was larger in the DCP than in the SCP [19] and the superficial FAZ area was significantly smaller for people over 60 years old [19]. Moreover, there were reported greater density values in the superficial vascular network in the foveal region, compared to the deep vascular network, with a smaller FAZ size at the superficial level [20]. In the parafoveal area, higher values of the VD were noticed in the deep vascular network [20].

## 5. Quantitative OCTA Assessment in Diabetic Macular Edema

There are various studies in the literature that investigated the modifications in the quantitative OCTA parameters in patients with DME.

### 5.1. Vessel Density

Vessel density (VD) is a parameter defined by the total length of perfused blood vessels per unit area in the region of measurement [21,22]. This metric is obtained after skeletonization [15]. For its calculation, each vessel, regardless of its caliber, is reduced to only one pixel-line, so to remove the effect of the vessel diameter from the analysis of the microcirculation [15]. It was reported that vessel density metrics were significantly lower in eyes with type 2 diabetes and DME when compared to patients with no diabetes [22]. The results of the same study showed lower central VD in patients with DME compared to healthy subjects [22] and also significantly lower inner VD and full VD in patients with DME compared to diabetic patients without diabetic retinopathy and patients with no diabetes [22]. Moreover, in patients with diabetic retinopathy, a lower VD of SCP, as well as a higher flow deficit of choriocapillaris was associated with an increased risk of DME development, independent of known risk factors [21].

Vessel skeletonized density (VSD) reduces the impact of large vessels on measurement and is considered to be more sensitive to retinal microvascular changes [15,23].

When comparing quantitative parameters between eyes with the same DR severity, in eyes with mild NPDR and DME, there was significantly lower skeleton density (SD) in the superficial retinal layer (SRL) and the deep retinal layer (DRL) compared to those without DME [15].

The measurement of OCTA metrics with a wider field swept-source OCTA device revealed that in mild NPDR associated with DME, there was a decrease in VSD in the DCP on 6 × 6 mm angiograms, whereas on 12 × 12 mm angiograms there was reduced VSD in the SCP and the full-thickness retina [23]. In the case of moderate-severe NPDR patients with DME, there was a significant reduction in VSD in the SCP and also of VSD in the full-thickness retina on 6 × 6 mm angiograms [23].

Regarding the influence of various treatment modalities on the quantitative parameters assessed by OCTA, in patients with DME treated with intravitreal anti-vascular endothelial growth factor (anti-VEGF), the first-line treatment for DME at the moment, one study concluded that the vessel length density (VLD)—based thresholding was more accurate for quantifying the retinal vascular changes after a single injection [24]. However, several studies showed no significant changes in this parameter after the anti-VEGF treatment [24].

Another study found that, at the second month of follow-up after an intravitreal dexamethasone implant (IDI), the vascular density in the perifoveal ring in the SCP on a 6 × 6 mm scan was significantly decreased [25]. Furthermore, in patients with previously treatment-naïve DME, after subthreshold micropulse yellow laser (SMYL) treatment, it resulted that there were no significant differences regarding the VD in the SCP, DCP and choriocapillary plexus (CCP) after the procedure [26].

Related to the correlations between visual acuity and quantitative OCTA microvascular parameters, in DME, it was noticed that low skeleton density in the DCP was correlated with poorer best-corrected visual acuity (BCVA), suggesting that macular ischemia plays an important role in the visual acuity drop associated with DME [27].

Table 1 summarizes the results of the studies analyzed, regarding the vessel density modifications related to DME.

### 5.2. Perfusion Density

Perfusion density, defined as the total area of perfused blood vessels per unit area in the region of measurement, is another metric used to quantitatively assess DME by OCTA. Compared to vessel density as defined above, this parameter takes into consideration both the length and the diameter of the vessel for the analysis, therefore being better in estimating the real vascular density [28]. However, in situations when perfusion reduction coexists with vascular dilations, it can induce false results regarding the microvascular quantification [28].

In diabetic chronic cystoid macular edema, compared to non-diabetic subjects, the mean value of this parameter was reduced in both SCP and DCP [29]. However, there was a greater reduction in perfusion density in the SCP, compared to DCP [29]. The authors of the study suggested two possible explanations for these results: either a software overestimation in the DCP due to the projection artifacts or a false interpretation of the walls between the cystic cavities and the lipid deposits in the cystoid spaces, inducing a correlation signal [29]. The same study concluded that in the cases of spontaneous or therapeutic remission of macular edema, the mean values of this parameter did not show significant changes [29].

Another study concluded that the global capillary density index (CDI), an equivalent metric used for quantifying the area occupied by vessels, was lower in patients with DME compared to patients without DME, independent of the capillary plexus affected [30].

In patients with diabetic retinopathy, a lower PD of SCP was associated with an increased risk of DME development, independent of known risk factors [21]. When comparing quantitative parameters between eyes with the same DR severity, in eyes with mild NPDR and DME, there were significantly lower values of this metric in the superficial retinal layer (SRL) and the deep retinal layer (DRL) in the eyes with DME compared to those without DME [15]. The impact of DME on deep vessel densities (DVD) was also confirmed by another study which showed that patients with NPDR with DME presented with reduced DVD, compared with patients without DR and also compared with patients with NPDR but without DME [31]. The same study also showed that the level of ANGPTL4, a cytokine correlated with the presence of DME, was an influencing factor for DVD [31]. Moreover, it was reported that density perfusion metrics were significantly lower in eyes with DME compared to patients with no diabetes and that, in the DME group of patients, PD was significantly lower in the DCP compared to the non-DME group of patients [22]. Furthermore, the peripapillary vessel density, as well as the retinal superficial and deep vascular density, and also the choriocapillaris density in the foveal and parafoveal areas, were significantly reduced in a study that compared patients with moderate NPDR with DME, with healthy subjects [32].

DME development was also associated with lower values of this metric in the SCP, independent of established risk factors including age, duration of diabetes, HbA_1c_, mean arterial blood pressure and severity of DR at baseline [33]. However, the eyes with DME had also impaired perfusion at the level of DCP, even though the alterations of SCP were the ones linked to the DME formation [33].

The OCTA assessment with a wider field swept-source OCTA device revealed that, on 6 × 6 mm angiograms, in mild NPDR associated with DME, at the DCP level, there was a decrease in the total area of perfused vessels per unit area in the region of measurement [23]. In the case of moderate–severe NPDR patients with DME, on 6 × 6 mm angiograms, there was a significant reduction in this metric in the SCP and in the full-thickness retina [23].

Differences were also noted regarding the OCTA quantitative parameters in patients with DME of different grades of severity [34]. It was concluded that patients with severe DME presented lower perfusion in the DVP, compared to early and advanced DME [34].

It was also shown that treatment naïve diabetic eyes with moderate or severe NPDR and DME had significantly lower superficial and deep perfusion values compared to healthy eyes [35]. Moreover, the diabetic eyes with DME presented lower values at the level of the deep retinal plexus, with the macular perfusion being more impaired compared to the superficial one in patients with DME compared to eyes without DME [35]. Furthermore, in patients with DME, a lower visual acuity was associated with a lower vessel area density at the level of the superficial retinal plexus [35].

Related to the influence of various therapeutic approaches on the perfusion density measurements in patients with DME, when comparing responders to anti-VEGF treatment with non-responders, it was noticed that in both groups of patients, there were lower values in the DCP than in the SCP [36], but poor responder DME eyes exhibited lower vascular flow density in the DCP and a lower flow density in the total capillary plexus (TCP) [36]. However, there were no significant differences between the two groups in the SCP [36]. It was suggested that the integrity of the perifoveal DCP is associated with anti-VEGF treatment response and the study found that there was an outer plexiform layer disruption in SD-OCT corresponding with the non-flow area of the DCP in OCTA [36]. The poor response to anti-VEGF in patients with impaired vascular flow in the DCP was explained by several theories. The first one stressed the inability of anti-VEGF agents to diffuse through the protein-rich intraretinal cysts and reach the capillaries in the DCP [36]. On the other hand, given the ischemic alterations in the deep retina, there may be an abundance of VEGF expression at that level, which may limit the efficacy of anti-VEGF agents [36]. Moreover, due to the selective tightening effect of anti-VEGF agents on the endothelial junctions, inhibition of VEGF cannot restore absent or broken vessels [36]. Last, given the implication of DCP in removing the excess fluid from the retina, its reduced vascular density limits the fluid evacuation, despite the action of the anti-VEGF agents [36].

A study on patients treated with fluocinolone acetonide intravitreal implant showed an increase in the parafoveal and perifoveal SCP perfusion density at the 4-month follow-up [37]. The improvement was considered to be the consequence of the beneficial effect of corticosteroids on leukostasis, a process that facilitates retinal non-perfusion and vascular leakage in patients with diabetic retinopathy [37].

Regarding the intravitreal dexamethasone implant (IDI) for the treatment of DME, it has been shown that the SCP density and the DCP density in the foveal and parafoveal area did not modify significantly during the follow-up at 7 days, 30 days, 60 days, 90 days and 120 days after the IDI implantation, despite the significant reduction in the macular thickness [32]. This was considered to be mainly induced by the irreversible ischemic alterations that occur in the retina [32]. In the same study, the choriocapillary density tended to increase after treatment, but the explanation was that, in the setting of edema, the OCT signal could have been attenuated, this being the reason why the choriocapillaris could have seemed reduced before the dexamethasone implant [32]. On the other hand, another study found that, at the second month of follow-up after IDI, vascular perfusion in the perifoveal ring in the SCP on a 6x6 mm scan was significantly decreased, and also reported a reduction in the vascular perfusion in the perifoveal ring in the DCP at months 2 and 3, and parafoveal ring at month 2, suggesting the importance of permanent capillary occlusion in DME [25].

After subthreshold micropulse yellow laser (SMYL) treatment in patients with previously treatment-naïve DME, no significant differences were found regarding the PD in the SCP, DCP and choriocapillary plexus (CCP) [26]. In another study, that evaluated the effect of SMYL on OCTA quantitative parameters in cases with persistent DME after pars plana vitrectomy for tractional DME, it resulted that, at 3-month and 6-month follow-up, the parafoveal density in the SCP and DCP was significantly higher in the SMYL group compared to the patients with DME that were only observed and not treated [38]. Similarly, it was found that during the first month after subthreshold yellow pattern laser therapy for DME, there was an increase in the mean values of this metric in the DCP [39].

Table 2 summarizes the results of the studies analyzed in relation to perfusion density modifications in DME.

### 5.3. Vessel Diameter Index (VDI)

Vessel diameter index (VDI) is defined by the proportion of area occupied by vessels divided by the skeletonized density [27] and it refers to the average vascular caliber on OCTA images [15]. When comparing quantitative parameters between eyes with the same DR severity, in eyes with mild NPDR and DME, there was a significantly higher VDI in the DRL compared to those without DME [15] (Table 3). In cases of severe NPDR, the eyes with DME showed a significantly higher VDI in the DRL compared with the eyes without DME [15] (Table 3).

### 5.4. Foveal Avascular Zone (FAZ) Parameters

When studying angiopoietin-like levels in the aqueous humor of patients with DME, it resulted that the levels of ANGPTL4 and ANGPTL6 were significantly higher in patients with DME, and ANGPTL4 correlated positively with the FAZ perimeter [22].

The measurement of the OCTA metrics with a wider field swept-source OCTA device showed that in mild NPDR associated with DME, there was a decrease in FAZ circularity on 6 × 6 mm angiograms [23]. Furthermore, in the case of moderate-severe NPDR patients with DME, on 6 × 6 mm angiograms, there was a decrease in the FAZ area [23].

It was also concluded that patients with severe DME presented a significant increase in the acircularity index, compared to early and advanced DME [34].

Moreover, the treatment naïve diabetic eyes with moderate or severe NPDR and DME had a significantly larger FAZ area at the level of the SCP than the diabetic eyes without edema and also than healthy subjects [35]. Furthermore, in patients with DME, a lower visual acuity was associated with a larger FAZ at both the superficial and the deep retinal plexuses [35].

When comparing responders to anti-VEGF treatment with non-responders, it was noticed that the DME eyes had a larger FAZ in the DCP than in the SCP, in both groups of patients [36]. Poor responder DME eyes, however, exhibited a larger FAZ area in the DCP, whereas there were no significant differences between the two groups in the SCP [36].

After SMYL laser treatment in patients with previously treatment-naïve DME, it resulted that the FAZ did not change significantly in the SCP, but there was a significant reduction in it in the DCP at 6 months compared to the initial evaluation [26]. In another study that evaluated the effect of SMYL, in cases with persistent DME after pars plana vitrectomy for tractional DME, on OCTA quantitative parameters analysis, it resulted that the FAZ area was significantly smaller in the SMYL group at 6-month follow-up [38]. In another study, though, it was found that during the first month after subthreshold yellow pattern laser therapy for DME, the FAZ area did not change significantly after the laser treatment [39].

Table 4 summarizes the results of the studies analyzed regarding the FAZ parameters modifications related to DME.

### 5.5. Fractal Dimension

The fractal dimension characterizes the architecture of the vascular network and provides an index of the branching complexity of the capillary network [15]. When comparing quantitative parameters between eyes with the same DR severity, in eyes with mild NPDR and DME, there was a significantly lower fractal dimension (FD) in the superficial retinal layer (SRL) and the deep retinal layer (DRL) compared to those without DME [15] (Table 5).

Regarding the correlations between the visual acuity and the quantitative OCTA microvascular parameters, in DME, a low FD in the DCP was correlated with a poorer BCVA, suggesting that macular ischemia plays an important role in the visual acuity drop associated with DME [27] (Table 5).

## 6. Deep Vascular Plexus Implications in Diabetic Macular Edema

The OCTA quantitative assessment of patients with diabetes mellitus has been a subject of interest, in the attempt to identify the starting point of the retinal microvascular impairment. Studies have shown that in patients with type 1 diabetes mellitus with no or mild signs of DR, there is reduced parafoveal vessel density in the DCP, compared to non-diabetic subjects [40] suggesting that DCP would be the first retinal vascular layer affected in diabetic patients. The early involvement of the DCP in the retinal ischemic process was also supported by the significant enlargement of FAZ noted in the DCP compared to SCP [41].

Regarding diabetic macular edema, most investigations are in agreement with the fact that, even though abnormalities of the superficial vascular plexus (SVP) have been identified, DME is mostly associated with decreased or absent vascular flow in the deep vascular plexus (DVP) as well. Many of the studies included in this review concluded that, at the level of the deep capillary plexus, in patients with DME, there was a significant decrease in the parameters used for the assessment of macular ischemia, such as vessel density, perfusion density, as well as FAZ enlargement. Several theories that explain this association have been proposed. It was stated that under normal circumstances, the fluid is produced at the level of the superficial vascular layer, and is absorbed, with the help of Müller cells, at the deep capillary plexus [42]. Macular edema starts to develop when there is an imbalance between fluid accumulation into the retina and its egress out of the retina [42]. The low or absent flow in the deep capillary plexus might be translated into an incapacity of the deep plexus to remove the fluid from the retina, resulting in the preferential development and the recurrence of cystoid spaces in the regions with impaired vascular flow [42]. On the other hand, the DVP itself might be the source of vascular leakage, leading to fluid accumulation in the deeper parts of the retina [43]. Moreover, compared to SVP, there is an abundance of very small vessels at the DVP level, increasing its susceptibility to leukostasis [30,42] and also its vulnerability to the VEGF effects [30]. Further investigations need to be made in order to fully explain the OCTA metrics changes in different plexuses in the setting of DME.

## 7. Sources of Errors in the Quantitative OCTA Assessment of Diabetic Macular Edema

One of the most important errors in the OCTA quantitative analysis is that of segmentation. In cases of retinal edema, there is an increase in the retinal layers’ thickness, and a decrease in the contrast between the retinal layers, therefore the segmentation algorithms that are based on a healthy retina cannot be reliable, generating errors [43]. The images with segmentation errors need a manual correction, the accuracy of the segmentation being in this case operator-dependent. After the correction of segmentation errors in diabetic patients with DME, one study showed that there was a change in the foveal and parafoveal vessel density measurements in both SCP and DCP, the change being greater in DCP [44]. Moreover, after segmentation error correction in patients with DME, a statistically significant difference was noted regarding the whole image and the parafoveal metrics assessment after the inner plexiform layer correction [45].

Another suggested cause of error in the quantitative OCTA evaluation of DME is the presence of cystoid spaces, that could exert a mechanical effect on the vessels, displacing them and falsely creating the impression of non-perfusion in the areas of the cysts [46]. This idea was supported by the fact that, after anti-VEGF treatment, a reappearance of vascularization was noticed in the area where the edema was previously identified [29]. However, according to other sources, no reperfusion was observed in the areas of the cysts after the fluid resolution, assuming that cystoid macular edema has a tendency to develop in areas of poor capillary perfusion [29,42]. Further studies are necessary to clarify this aspect.

One more potential source of error regarding the OCTA metrics assessment is the axial length. It was reported that, for healthy eyes with long axial lengths, in order to obtain an accurate quantification of the superficial retinal vessel density and of the foveal avascular zone area, correction formulae should be applied to compensate for the image magnification [47]. Magnification influences how much of the FAZ is included in the calculation and also the perfusion density, this being an important aspect in myopic eyes [48].

Signal strength is another essential element that can influence the OCTA measurements, with images with higher signal strength providing better clarity and improved segmentation and leading to increased accuracy of the measurement, whereas lower signal strength could lead to an underestimation of the vascular quantitative parameters [49].

## 8. Conclusions

Quantitative optical coherence tomography angiography parameters represent promising instruments that can aid in the diagnosis, therapeutic decisions, follow-up and anatomic and functional prognostic in patients with diabetic macular edema. The standardization of nomenclature and the development of new strategies that can offer the possibility of data comparison between different devices represent some of the further directions that need to be addressed for a better understanding of the utility of OCTA metrics in clinical practice.

## Figures and Tables

**Table 1 medicina-59-01120-t001:** Vessel density (defined by the total length of perfused blood vessels per unit area in the region of measurement) in Diabetic Macular Edema.

Study Details	Study Design	Type of Diabetes	Study Groups	OCTA Software	Area of Scan	Results
Yan et al. (2021) [22]	Cross-sectional	2	Non-diabetic patients (control group) Diabetic patients categorized into the NDR group and DME group	AngioPlex (Software version CIRRUS 11.0, Carl Zeiss Meditec)	3 mm × 3 mm	Lower central VD in the DME group compared to control group. Significantly lower inner VD and full VD in the DME group, compared to NDR and control group.
Guo et al. (2022) [21]	Prospective cohort study	2	Diabetic patients without DR (NDR) or with mild NPDR and 35-80 years old at baseline, without any DME and without a history of ocular therapy	DRI OCT Triton (Software version n.a.)	3 mm × 3 mm centered on the ONH	A lower VD of SCP, as well as a higher flow deficit of choriocapillaris were associated with an increased risk of DME development, independent of known risk factors.
Kim et al. (2016) [15]	Retrospective, cross-sectional, observational	n.a.	Eyes with diabetic retinopathy Healthy eyes	n.a.	3 mm × 3 mm	Significantly lower skeleton density (SD) in eyes with mild NPDR and DME in the superficial retinal layer (SRL) and the deep retinal layer (DRL) compared to those without DME.
Garg et al. (2022) [23]	Cross-sectional Observational	1, 2	Diabetic patients Control group	PLEX Elite 9000 (Macular Density v0.7.3 on the ARI Network, Zeiss Portal v.5.4-1206)	6 mm × 6 mm 12 mm × 12 mm	A decrease in VSD in the DCP on 6 × 6 mm angiograms in mild NPDR associated with DME. Reduced VSD in the SCP and the full-thickness retina on 12 × 12 mm angiograms. Significant reduction in VSD in the SCP and also in the full-thickness retina on 6 × 6 mm angiograms in the case of moderate–severe NPDR patients with DME.
Song et al. (2022) [24]	Prospective	1, 2	Diabetic patients with central-involving DME	AngioVue (Software version 2017.1.0.151)	3 mm × 3 mm	The VLD-based thresholding was more accurate for quantifying the retinal vascular changes after a single injection in patients with DME treated with intravitreal anti-vascular endothelial growth factor (anti-VEGF). No significant changes in VLD were identified in the short term after the treatment.
Carnota-Méndez et al. (2022) [25]	Prospective, non-randomized, open-label	1, 2	Type 1 or 2 Diabetes and Center-involving DME patients who underwent intravitreal dexamethasone implant (IDI)	AngioPlex (Software version n.a.)	6 mm × 6 mm	The vascular density in the perifoveal ring in the SCP at the second month of follow-up after intravitreal dexamethasone implant (IDI), was significantly decreased.
Vujosevic et al. (2020) [26]	Prospective, longitudinal, case-series	2	Treatment-naïve patients with DME treated with SMPL	DRI OCT Triton plus (Software version 10.07.003.03)	3 mm × 3 mm	No significant differences regarding the VD in the SCP, DCP and choriocapillary plexus (CCP) at 3 months and 6 months after the laser procedure.
Hsiao et al. (2020) [27]	Retrospective Cross-sectional	2	Type 2 Diabetic patients with DMO	AngioVue (Software version 201.2.0.93)	3 mm × 3 mm	Low skeleton density in the DCP was correlated with poorer best-corrected visual acuity (BCVA).

**Table 2 medicina-59-01120-t002:** Perfusion density (defined as the total area of perfused blood vessels per unit area in the region of measurement) in Diabetic Macular Edema.

Study Details	Study Design	Type of Diabetes	Study Groups	OCTA Software	Area of Scan	Results
Mané et al. (2016) [29]	Retrospective	1, 2	Patients with DR associated with chronic diabetic cystoid macular edema	AngioVue (Software version 2015.1.0.71)	3 mm × 3 mm	The mean value of PD was reduced in both SCP and DCP in diabetic chronic cystoid macular edema, compared to non-diabetic subjects. There was a greater reduction in perfusion density in the SCP, compared to DCP. The mean values of PD did not show significant changes in the cases of spontaneous or therapeutic remission of macular edema.
Ting et al. (2017) [30]	Prospective Observational	2	Patients with type 2 diabetes, with and without DR	n.a.	3 mm × 3 mm	The global capillary density index (CDI) was lower in patients with DME compared to patients without DME, independent of the capillary plexus affected.
Guo et al. (2022) [21]	Prospective cohort study	2	Diabetic patients without DR (NDR) or with mild NPDR and 35-80 years old at baseline Subjects without any DME Subjects without a history of ocular therapy	DRI OCT Triton (Software version n.a.)	3 mm × 3 mm centered on the ONH	In patients with diabetic retinopathy, a lower PD of SCP was associated with an increased risk of DME development, independent of known risk factors.
Kim et al. (2016) [15]	Retrospective, cross-sectional, observational	n.a.	Eyes with diabetic retinopathy and healthy eyes	n.a.	3 mm × 3 mm	Significantly lower values of PD in the superficial retinal layer (SRL) and the deep retinal layer (DRL) in the eyes with mild NPDR and DME compared to those without DME.
Xu et al. (2022) [31]		2	NPDR patients with DME, Senile cataract patients with type 2 diabetes mellitus who received phacoemulsification diagnosed as NPDR without DME Diabetic patients without retinopathy (control group)	AngioVue (Software version n.a.)	3 mm × 3 mm	Patients with NPDR with DME presented with reduced DVD, compared with patients without DR and also compared with patients with NPDR but without DME.
Yan et al. (2021) [22]	Cross-sectional	2	Non-diabetic patients (control group) Diabetic patients categorized into the NDR group and DME group	AngioPlex (Software version CIRRUS 11.0, Carl Zeiss Meditec)	3 mm × 3 mm	PD was significantly lower in eyes with DME compared to patients with no diabetes. PD was significantly lower in the DCP in the DME group of patients compared to non-DME group of patients.
Toto et al. (2017) [32]	Prospective	2	Type 2 diabetic patients with moderate stage DR and treatment naïve DME, treated with IDI Control group—healthy patients	AngioVue (Software version 2015.1.0.90)	3 mm × 3 mm	The peripapillary vessel density, the retinal superficial and deep vascular density and the choriocapillaris density in the foveal and parafoveal areas, were significantly reduced. The SCP and the DCP density in the foveal and parafoveal area did not modify significantly during the follow-up at 7, 30 days, 60 days, 90 days and 120 day after the IDI implantation, despite the significant reduction in the macular thickness. The choriocapillary density tended to increase after treatment.
Sun et.al (2019) [33]	Prospective Observational	1, 2	Patients with diabetes mellitus	DRI OCT Triton (Software IMAGEnet6 v.1.23.15008, Basic License 10)	3 mm × 3 mm	DME development was associated with lower values of PD in the SCP, independent of established risk factors including age, duration of diabetes, HbA_1c_, mean arterial blood pressure and severity of DR at baseline. The eyes with DME had also impaired perfusion at the level of DCP, even though the alterations of SCP were the ones linked to the DME formation.
Garg et al. (2022) [23]	Cross-sectional Observational	1, 2	Diabetic patients Control group	PLEX Elite 9000 (Macular Density v0.7.3 on the ARI Network, Zeiss Portal v.5.4-1206)	6 mm × 6 mm 12 mm × 12 mm	A decrease in PD in mild NPDR associated with DME, at the DCP level, on 6 × 6 mm angiograms. A significant reduction in PD in the SCP and in the full-thickness retina in moderate-severe NPDR patients with DME, on 6 × 6 mm angiograms.
Han et al. (2022) [34]	Cross-sectional Observational	2	Patients diagnosed with early, advanced and severe DME	AngioVue (Algorithm version A2017, 1, 0, 155)	3 mm × 3 mm	Lower perfusion in the DVP for patients with severe DME, compared to early and advanced DME.
AttaAllah et al. (2019) [35]	Observational case series	n.a.	Diabetic patients with DME Diabetic patients without DME Healthy individuals (control group)	AngioVue (RTVue-XR version 2017.1.0.151)	6 mm × 6 mm	Treatment naïve diabetic eyes with moderate or severe NPDR and DME had significantly lower superficial and deep perfusion values compared to healthy eyes. The diabetic eyes with DME presented lower values at the level of the deep retinal plexus, compared to eyes without DME. In patients with DME, a lower visual acuity was associated with a lower vessel area density at the level of the superficial retinal plexus.
Lee et al. (2016) [36]	Retrospective case-control study	2	Patients with type 2 diabetes, with any stage of DR Cases—DME eyes poor responders to anti-VEGF agents Controls—age-matched DME eyes with good response to anti-VEGF agents Fellow eyes without DME for the cases and controls	AngioVue (Software version n.a.)	3 mm × 3 mm	When comparing responders to anti-VEGF treatment with non-responders, it was noticed that in both groups of patients there were lower values in the DCP than in the SCP Poor responder DME eyes exhibited lower vascular flow density in the DCP and a lower flow density in the total capillary plexus (TCP). There were no significant differences between the 2 groups in the SCP.
Brambati et al. (2022) [37]	Retrospective cohort study	2	Patients with type 2 NPDR and DME	PLEX Elite 9000 (Software version n.a.)	6 mm × 6 mm	An increase in the parafoveal and perifoveal SCP perfusion density at the 4-months follow-up.
Carnota-Méndez et al. (2022) [25]	Prospective, non-randomized, open-label	1,2	Type 1 or 2 Diabetes and Center-involving DME patients who underwent intravitreal dexamethasone implant (IDI)	AngioPlex (Software version n.a.)	6 mm × 6 mm	A significant decrease in the vascular perfusion in the perifoveal ring in the SCP on a 6 × 6 mm scan, at the second month of follow-up after IDI A reduction in the vascular perfusion in the perifoveal ring in the DCP at months 2 and 3 A reduction in the vascular perfusion in the parafoveal ring at month 2.
Vujosevic et al. (2020) [26]	Prospective, longitudinal, case-series	2	Treatment-naïve patients with DME treated with SMPL	DRI OCT Triton plus (Software version 10.07.003.03)	3 mm × 3 mm	After subthreshold micropulse yellow (SMYL) laser treatment no significant differences were found regarding the PD in the SCP, DCP and choriocapillary plexus (CCP).
Bonfiglio et al. (2022) [38]	Prospective, comparative, non-randomized, pilot study	2	Pseudophakic patients with a persistent DME after PPV for tractional DME, are divided into two groups: DME eyes who received micropulse subthreshold laser treatment (SMYL group) DME eyes observed after PPV without treatment (control group)	AngioVue (Software version 2017.1.0.151 AngioVue Phase 7 software with PAR)	6 mm × 6 mm	At 3-month and 6-month follow-ups, the parafoveal density in the SCP and DCP was significantly higher in the SMYL group compared to the patients with DME that were only observed and not treated.
Karaca et al. (2022) [39]	Prospective	2	Diabetic patients diagnosed as naïve DME with DR (both non-proliferative and proliferative types)	AngioVue (Software version n.a.)	6 mm × 6 mm	An increase in the mean values of PD in the DCP, during the first month after subthreshold yellow pattern laser therapy for DME.

**Table 3 medicina-59-01120-t003:** Vessel diameter index (defined by the proportion of area occupied by vessels divided by the skeletonized density) in Diabetic Macular Edema.

Study Details	Study Design	Type of Diabetes	Study Groups	OCTA Software	Area of Scan	Results
Kim et al. (2016) [15]	Retrospective, cross-sectional, observational	n.a.	Eyes with diabetic retinopathy and healthy eyes	n.a.	3 mm × 3 mm centered on the fovea	Significantly higher VDI in the DRL in eyes with mild NPDR and DME, compared to those without DME. In cases of severe NPDR, the eyes with DME showed a significantly higher VDI in the DRL compared with the eyes without DME.

**Table 4 medicina-59-01120-t004:** FAZ parameters modifications related to Diabetic Macular Edema.

Study Details	Study Design	Type of Diabetes	Study Groups	OCTA Software	Area of Scan	Results
Garg et al. (2022) [23]	Cross-sectional Observational	1, 2	Diabetic patients Control group	PLEX Elite 9000 (Macular Density v0.7.3 on the ARI Network, Zeiss Portal v.5.4-1206)	6 mm × 6 mm 12 mm × 12 mm	A decrease in FAZ circularity on 6 × 6 mm angiograms in mild NPDR associated with DME. A decrease in the FAZ area in the case of moderate-severe NPDR patients with DME, on 6 × 6 mm angiograms.
Han et al. (2022) [34]	Cross-sectional Observational	1, 2	Patients diagnosed with early, advanced and severe DME	AngioVue (Algorithm version A2017, 1, 0, 155)	3 mm × 3 mm	A significant increase in the acircularity index for patients with severe DME compared to early and advanced DME.
AttaAllah et al. (2019) [35]	Observational case series	n.a.	Diabetic patients with DME Diabetic patients without DME Healthy individuals (control group)	AngioVue (RTVue-XR version 2017.1.0.151)	6 mm × 6 mm	Significantly larger FAZ area at the level of the SCP in the case of the treatment-naïve diabetic eyes with moderate or severe NPDR and DME compared to diabetic eyes without edema and also to healthy subjects. A lower visual acuity was associated with a larger FAZ at both the superficial and the deep retinal plexuses in patients with DME.
Lee et al. (2016) [36]	Retrospective case-control study	2	Patients with type 2 diabetes, with any stage of DR Cases—DME eyes poor responders to anti-VEGF agents Controls—age-matched DME eyes with good response to anti-VEGF agents Fellow eyes without DME for the cases and controls	AngioVue (Software version n.a.)	3 mm × 3 mm	When comparing responders to anti-VEGF treatment with non-responders, it was noticed that the DME eyes had a larger FAZ in the DCP than in the SCP, in both groups of patients. Poor responder DME eyes exhibited a larger FAZ area in the DCP PNo significant differences between the 2 groups in the SCP.
Vujosevic et al. (2020) [26]	Prospective, longitudinal, case-series	2	Treatment-naïve patients with DME treated with SMYL	DRI OCT Triton plus (Software version 10.07.003.03)	3 mm × 3 mm	After SMYL laser treatment it resulted that the FAZ did not change significantly in the SCP, but there was a significant reduction in it in the DCP at 6 months compared to the initial evaluation.
Bonfiglio et al. (2022) [38]	Prospective, comparative, non-randomized, pilot study	2	Pseudophakic patients with a persistent DME after PPV for tractional DME, divided into two groups: DME eyes who received micropulse subthreshold laser treatment (SMYL group) DME eyes observed after PPV without treatment (control group)	AngioVue (Software version 2017.1.0.151 AngioVue Phase 7 software with PAR)	6 mm × 6 mm	The FAZ area was significantly smaller in the SMYL group at 6-month follow-up.
Karaca et al. (2022) [39]	Prospective	2	Diabetic patients diagnosed as naïve DME with DR (both non-proliferative and proliferative types)	AngioVue (Software version n.a.)	6 mm × 6 mm	During the first month after subthreshold yellow pattern laser therapy for DME, the FAZ area did not change significantly after the laser treatment.

**Table 5 medicina-59-01120-t005:** Fractal dimension in Diabetic Macular Edema.

Study Details	Study Design	Type of Diabetes	Study Groups	OCTA Software	Area of Scan	Results
Kim et al. (2016) [15]	Retrospective, cross-sectional, observational	n.a.	Eyes with diabetic retinopathy and healthy eyes	n.a.	3 mm × 3 mm centered on the fovea	Significantly lower FD in the superficial retinal layer (SRL) and the deep retinal layer (DRL) in eyes with mild NPDR and DME, compared to those without DME
Hsiao et al. (2020) [27]	Retrospective Cross-sectional	2	Type 2 Diabetic patients with DMO	AngioVue (Software version 201.2.0.93)	3 mm × 3 mm	A low FD in the DCP was correlated with a poorer BCVA

## Data Availability

Not applicable.

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
