# Peer review of "Quantitative Parameters Relevant for Diabetic Macular Edema Evaluation by Optical Coherence Tomography Angiography"

_medicina, 2023, doi:10.3390/medicina59061120_

Round 1

Reviewer 1 Report (New Reviewer)

Authors reviewed the quantitative OCTA parameters in DME and documented the changes in main perfusion and nonperfusion metrics in the superficial and deep and total retinal layers. The specific comments are below.

1.      They may discuss the impacts of segmentation error in the deep OCTA slab in eyes with DME.

2.      Lines 78-80  The sentence is not consistent to the inner retinal atrophy after retinal artery occlusion.

3.      Lines 189-192  Anti-VEGF treatment is the first line modality for DME in the global setting, and they may incorporate the statements regarding it.

4.      They prepared tables regarding vessel density and perfusion density. They may explain the differences between these parameters more clearly.

5.      The items in Tables are not appropriate, and they may improve them.

6.      The contents of Table 4 seem to be redundant, and can be incorporated into Tables 1 and 2.

7.      They stated the feasibility of OCTA parameters in DME diagnosis, although typical diagnostic criteria, i.e., CSME and CIDME, are dependent on fundus examination and OCT, respectively.

Author Response

Dear Reviewer,

Thank You for your comments and suggestions.

We’ve tried to answer point by point to each item.

  1. They may discuss the impacts of segmentation error in the deep OCTA slab in eyes with DME.

We brought into discussion the impacts of segmentation error in the deep OCTA slab in eyes with DME, as suggested (lines 477-483).

  1. Lines 78-80 The sentence is not consistent to the inner retinal atrophy after retinal artery occlusion.

The anatomical distribution of the retinal vascularization as detailed in lines 78-80 is unanimously accepted. The statement is consistent with the immediate aspect of the retina, in the edematous phase of the central retinal artery occlusion.

  1. Lines 189-192  Anti-VEGF treatment is the first line modality for DME in the global setting, and they may incorporate the statements regarding it.

We incorporated additional information regarding the impact of anti-VEGF treatment on vessel length density (lines 216-222). 

  1. They prepared tables regarding vessel density and perfusion density. They may explain the differences between these parameters more clearly.

We added further explanations regarding the differences between vessel density and perfusion density and their role in the microvascular quantification by means of OCTA.

We emphasized the fact that vessel density evaluates the retinal microvascular abnormalities without taking into consideration the vessel diameter, therefore being more sensitive to the perfusion changes at the capillary level (lines 189-192, lines 201-203), while perfusion density takes into consideration both the length and diameter of the vessel, therefore providing the best estimate of the real vascular density (lines 243-246).

However, we mentioned that one potential disadvantage of perfusion density is that it doesn’t take into consideration the situations when the perfusion reduction coexists with vascular dilations, inducing false results in the microvascular quantification (lines 246-248).

  1. The items in Tables are not appropriate, and they may improve them.

We brought two more columns in each Table, with additional specifications about the studies mentioned: the type of diabetes in the studied population and the OCTA software used for patient assessment.

  1. The contents of Table 4 seem to be redundant, and can be incorporated into Tables 1 and 2.

While we are very thankful for this suggestion, we mention that we chose this modality of presenting the information in order to make it more visible and easier to read.

Table 4 contains information regarding FAZ modifications in the setting of diabetic macular edema, while Tables 1 and 2 contain information regarding two other different vascular parameters: vessel density and perfusion density, respectively.

We do not feel that by mixing the information in Table 4 with the ones in Tables 1 and 2 we would make the information more efficiently transmitted. 

  1. They stated the feasibility of OCTA parameters in DME diagnosis, although typical diagnostic criteria, i.e., CSME and CIDME, are dependent on fundus examination and OCT, respectively.

The purpose of our review was not to claim that OCTA parameters might substitute the standard diagnostic modalities of DME.

With our research, we wanted to emphasize that by quantifying the degree of ischemia associated with fluid accumulation in the macula, OCTA metrics could have an important role in the assessment of patients with DME and be useful adjuvants to the gold standard modalities of DME diagnosis, treatment and monitoring.  

Reviewer 2 Report (New Reviewer)

The manuscript entitled “Quantitative parameters relevant for diabetic macular edema evaluation by the optical coherence tomography angiography” by Lazar and colleagues reviewed OCTA findings on diabetic retinopathy and diabetic macular edema. As the authors concluded, OCTA is exciting imaging tool to evaluate diabetic changes in retina and choroid, and the review is well written. There are several concerns as below.

1. The authors focus on different parameters of OCTA, so the status of diabetic retinopathy is disparate. For example, some eyes have no retinopathy, some have macular edema, and some have pre- and post-treatment conditions, which causes confusion for the reader. For example, it would be easier to understand if the information were divided into NDR and DR, presence or absence of DME, and pre- and post-treatment changes in DME.

2. Type 2 diabetes has many complications that affect OCTA parameters, such as hypertension and hyperlipidemia. It may be necessary to consider this point as well.

3. In addition to the comment in 2), since type 1 DM has a clear point of onset and many cases have few systemic complications, why not focus on what is known about patients with type 1 DM (see references below)?

Niestrata-Ortiz M, Fichna P, Stankiewicz W, Stopa M. Enlargement of the foveal avascular zone detected by optical coherence tomography angiography in diabetic children without diabetic retinopathy. Graefes Arch Clin Exp Ophthalmol. 2019;257(4):689-697.

Simonett JM, Scarinci F, Picconi F, et al. Early microvascular retinal changes in optical coherence tomography angiography in patients with type 1 diabetes mellitus. Acta Ophthalmol. 2017;95(8):e751-e755.

4. There are countless more OCTA reports on diabetes, so please clarify from what perspective you have decided to cite the references.

Author Response

Dear Reviewer,

Thank You for your comments and suggestions.

We’ve tried to answer point by point to each item.

  1. The authors focus on different parameters of OCTA, so the status of diabetic retinopathy is disparate. For example, some eyes have no retinopathy, some have macular edema, and some have pre- and post-treatment conditions, which causes confusion for the reader. For example, it would be easier to understand if the information were divided into NDR and DR, presence or absence of DME, and pre- and post-treatment changes in DME.

The purpose of our review, as stated right from the title, was to focus primarily on the OCTA quantitative assessment of patients with DME as a target population and not on patients with DR as a central point of interest.

Our main goal was to document the OCTA metrics modifications in patients with DME.

However, in our analysis, we took into consideration studies that investigated the relationship between DR and DME and we documented their results also. By doing this, the main purpose was to show the influence of macular ischemia in the development of DME at patients that already have DR, as well as the impact of the co-existence of both pathologies on the OCTA metrics modifications.

  1. Type 2 diabetes has many complications that affect OCTA parameters, such as hypertension and hyperlipidemia. It may be necessary to consider this point as well.

We are in agreement with your perspective, as studies show that systemic factors such as hypertension and hyperlipidemia may influence OCTA parameters as well. However, many of the studies analyzed for this review didn’t focus on investigating the influence of the co-existence of these systemic factors and DME on OCTA metrics, or they acknowledged that the OCTA parameters’ modifications were independent of other systemic factors associated.

Being an interesting point to consider, but also the subject of extensive discussion, the possible influence of systemic factors in the quantitative OCTA assessment of patients with DME remains to be the subject of further research in a different paper.

  1. In addition to the comment in 2), since type 1 DM has a clear point of onset and many cases have few systemic complications, why not focus on what is known about patients with type 1 DM (see references below)?

Niestrata-Ortiz M, Fichna P, Stankiewicz W, Stopa M. Enlargement of the foveal avascular zone detected by optical coherence tomography angiography in diabetic children without diabetic retinopathy. Graefes Arch Clin Exp Ophthalmol. 2019;257(4):689-697.

Simonett JM, Scarinci F, Picconi F, et al. Early microvascular retinal changes in optical coherence tomography angiography in patients with type 1 diabetes mellitus. Acta Ophthalmol. 2017;95(8):e751-e755.

We found the suggestion to be really interesting, bringing a different perspective on the OCTA quantitative parameters’ modifications in diabetic patients.

As it helped us emphasize what we had already stated in our article regarding the OCTA metrics’ modifications in the DCP in patients with DME, we also brought into discussion the OCTA metrics modifications found in patients with type 1 DM with no or mild signs of DR that highlight the early involvement of the DCP in the retinal ischemic process (lines 437-445).  

  1. There are countless more OCTA reports on diabetes, so please clarify from what perspective you have decided to cite the references.

While doing the research, our attention was mainly focused on the OCTA quantitative assessment of diabetic patients in relation to diabetic macular edema.

As such, the citations included relevant studies that investigated OCTA quantitative parameters’ modifications that illustrated the status of macular perfusion in the setting of DME.

Round 2

Reviewer 1 Report (New Reviewer)

All concerns are addressed.

This manuscript is a resubmission of an earlier submission. The following is a list of the peer review reports and author responses from that submission.

Round 1

Reviewer 1 Report

In the present review, Lazăr AS and co-workers try to summarize current knowledge on OCTA quantitative evaluation and limits in DME.

Paragraphs 2 to 4 focus on current quantitative assessments provided by analyses of OCTA images. However there is plenty of papers/reviews describing these aspects, so I suggest to remove/significantly reduce this initial part of the paper, as it does not add anything new to many previous works on the topic.

Paragraphs on OCTA in DME could be interesting, however they are very difficult to read and should be better organized, e.g. subdividing paragraphs based on single OCTA parameters (VD/PD/FAZ metrics and so on).

Limits of quantitative analyses performed on OCTA in DME should be stressed with more strength as results could be significantly altered due to the presence of artifacts. Another source of artifacts should be added in the relative paragraph, which is artifacts determined by the presence of intraretinal cysts, which can displace retinal vessels creating false areas of non-perfusion in the image (see https://doi.org/10.3928/23258 160-20160126-02; https://doi.org/10.1007/s00592-019-01424-4).

Author Response

Dear Reviewer,

Thank You for your suggestions. We seriously took them in consideration in order to prepare a better article.

We'll reply point by point:

  1. Regarding the paragraphs 2 to 4, they were intended to create an overview regarding the OCTA assessment of the retinal vasculature, bringing into light a series of theoretical and practical aspects that would aid in the understanding of the rest of the paper. Their purpose is to create the background for a more detailed discussion about the quantitative OCTA evaluation of the retinal vessels in DME.

We reduced the paragraph 2 (the one related to OCTA microvascular evaluation of the retina) to the most significant aspects regarding the OCTA assessment of the retinal vascular plexuses.

The paragraph 3 (related to the process of OCTA metrics calculation) encompasses a basic overview, intended to draw attention on the importance of the critical evaluation of the current studies, especially related to terminology, taking into consideration the heterogeneity of the OCTA devices, software and algorithms used in clinical practice to calculate all the parameters discussed.

Furthermore, the paragraph 4 (related to the quantitative OCTA assessment in healthy individuals) brings into attention the physiologic aspects of OCTA metrics evaluation in healthy eyes. Related to the whole context of the paper, we wanted to emphasize that even though these parameters can present with normal variations related to the retinal plexus evaluated, age and sex, however, when comparing healthy individuals with patients with DME, it resulted that the alterations were much more important in the DME category of individuals. The paragraph strengthens the results reported later on, regarding differences in the OCTA metrics between normal eyes and eyes with DME.  

  1. Related to OCTA in DME, we followed your recommendation regarding the subdivision of the paragraphs based on single OCTA parameters. We expect that the paper would now be easier to read and the message of the paper would be more efficiently delivered.
  2. Regarding the limits of the quantitative analyses performed on OCTA in DME, we added the source of artifacts you suggested, related to the displacement of the retinal vessels by the intraretinal cysts, which could falsely induce the appearance of areas of non-perfusion in the image. However, the main reason this was not mentioned in the first version of the manuscript, is the contradiction found in the literature. While some sources assume that the cystoid spaces exert a mechanical effect on the vessels, laterally displacing them and leading to false results of non-perfusion, some other sources suppose that there is a direct link between the development of cystoid macular edema and capillary dropout, with a tendency of the edema to form in areas of poor perfusion (DOI: 10.1097/IAE.0000000000001289 ; DOI: 10.1097/IAE.0000000000001158).

We think further studies are necessary for a solid conclusion in this regard.

We'll also add the revised article as an attachment, so "please see also the attachment"

Thank You!

Best regards!

Reviewer 2 Report

Diabetic macular edema is the main complication associated with diabetic retinopathy that causes blurred vision. The main topic of this review addressed by research is to detect DME through an innovative novel non-invasive imaging optical coherence tomography angiography, OCTA. In my opinion, the issue is scientifically relevant and interesting for the readers. The text is well-written. The methodology by which the review is conducted is accurate. Conclusions are consistent with the main topic. All references are appropriate.

Author Response

Dear Reviewer,

Thank You very much for your support.

We'll attach the final version of our article, so "please see the attachment" if you would like to do it.

Best regards!
